SCIENCE FORUM

# A community-led initiative for training in reproducible research

**Abstract** Open and reproducible research practices increase the reusability and impact of scientific research. The reproducibility of research results is influenced by many factors, most of which can be addressed by improved education and training. Here we describe how workshops developed by the Reproducibility for Everyone (R4E) initiative can be customized to provide researchers at all career stages and across most disciplines with education and training in reproducible research practices. The R4E initiative, which is led by volunteers, has reached more than 3000 researchers worldwide to date, and all workshop materials, including accompanying resources, are available under a CC-BY 4.0 license at https://www.repro4everyone.org/.

**SUSANN AUER[†], NELE A HAELTERMANN[†], TRACEY L WEISSGERBER, JEFFREY C ERLICH, DAMAR SUSILARADEYA, MAGDALENA JULKOWSKA, MAŁGORZATA ANNA GAZDA, BENJAMIN SCHWESSINGER*, NAFISA M JADAVJI* AND REPRODUCIBILITY FOR EVERYONE TEAM**

**\*For correspondence:**
benjamin.schwessinger@anu.edu.au (BS); njadav@midwestern.edu (NMJ)

[†]These authors contributed equally to this work

**Competing interests:** The authors declare that no competing interests exist.

## Why is training in reproducibility needed?

Reproducibility and replicability are central to science. Reproducibility is the ability to regenerate a result using the dataset and data analysis workflow that was used in the original study, while replicability is the ability to obtain similar results in a different experimental system (*Leek and Peng, 2015*; *Schloss, 2018*). Despite their importance, studies have shown that it can be quite challenging to reproduce and replicate peer-reviewed results (*Baker and Penny, 2016*; *Freedman et al., 2015*). In the past few years, several multi-center projects have assessed the level of reproducibility and replicability in various scientific fields, and have identified major factors that are critical for repeating and confirming scientific results (*Alsheikh-Ali et al., 2011*; *Amaral et al., 2019*; *Baker et al., 2014*; *Button et al., 2013*; *Cova et al., 2021*; *Errington et al., 2014*; *Friedl, 2019*; *Hardwicke et al., 2018*; *Lazic, 2010*; *Marqués et al., 2020*; *Open Science Collaboration, 2015*; *Shen et al., 2012*; *Stevens, 2017*; *Strasak et al., 2007*; *Weissgerber et al., 2019*; *Weissgerber et al., 2015*). In the rest of this article we will use the term reproducibility as shorthand for reproducibility and replicability, as is often done in the life sciences (*Barba, 2018*).

The factors that control the reproducibility of an experiment can be grouped into the four categories shown in *Figure 1*. The first represents technical factors, such as variability in reagents or materials used to perform research. The second category contains factors related to flaws in study design and/or statistical analysis such as the use of inappropriate controls, insufficient sample sizes to properly power the study, inappropriate statistical analyses, underpowered studies, and others. The third category contains human factors, which includes insufficient description of methods and the use of reagents or organisms that are not shared. In addition, scientific misconduct, such as hypothesizing after results have been obtained (HARKing; *Kerr, 1998*) or P-hacking (*Fraser et al., 2018*; *Head et al., 2015*; *Miyakawa, 2020*), is hard to detect and contributes to confirmation and

**Figure 1.** Factors that affect reproducibility in research. An approximation of the classification of categories that contribute to irreproducible scientific results, including technical, human, errors in study design and statistical analysis and external. Specific examples have been listed under each category.

publication bias issues. Lastly, external factors that are beyond the researchers' control can negatively impact reproducibility; these can include scientific rewards such as a high impact publication or paywalls that restrict access to crucial information. Going forward, developing solutions to minimize these confounding factors will be of vital importance to improve scientific integrity and to further accelerate the advancement of the scientific enterprise (*Botvinik-Nezer et al., 2020*; *Fomel and Claerbout, 2009*; *Friedl, 2019*; *Gentleman and Temple Lang, 2007*; *Mangul et al., 2019*; *Mesirov, 2010*; *NIH, 2020*; *Peng, 2011*).

While the problems with experimental reproducibility have been known for decades, they have only come to the fore over the past ten years (*Begley and Ellis, 2012*; *Munafò et al., 2017*; *Prinz et al., 2011*). Within the scientific community, systemic solutions and tools are being developed that allow scientists to efficiently share research materials, protocols, data, and computational analysis pipelines (some of these tools are covered in our training materials, see *Box 1*). Despite their transformative potential, these tools are underutilized, as most researchers are unaware of their existence, or do not know how to incorporate them in their daily workflows.

Integrating these tools into the standard scientific workflow has the potential to shift the scientific community towards a more transparent

and reproducible future. Educational initiatives with open-source materials can significantly increase the reach of teaching materials (*Lawrence et al., 2015*) to accelerate the uptake of best practices and existing tools for reproducible research. Several initiatives exist that offer tutorials or seminars on some aspects of reproducibility (*Box 2*). While they each have their strengths, none of them individually offer a scalable solution to the existing training gap in reproducibility. Here, we present Reproducibility for Everyone, a set of workshop materials and modules that can be used to train researchers in reproducible research practices. Our trainings are scalable, from a dozen attendees in an intensive workshop to a few hundred participants in an introductory workshop that can attend at once in a virtual format or a large venue. However, the reproducibility movement worldwide is growing, and as different initiatives cover various aspects of the training process, they can together help bridge the reproducible training gap.

## Reproducibility for Everyone (R4E)

R4E was formed in 2018 to address the challenges of integrating reproducible research practices in life science laboratories across the globe. Our mission is to increase awareness of the factors that affect reproducibility, and to promote best practices for reproducible and transparent scientific research. We offer open access introductory materials and workshops to teach scientists at all career stages and across disciplines about concrete steps they can take to improve the transparency and reproducibility of their research. All workshops are offered free of charge. We developed eight modules as independent, in-depth slide sets focusing on different aspects of the day-to-day scientific workflow, allowing trainers to customize the workshop and adapt it to audiences in different disciplines (*Box 1*). R4E targets mainly biological and medical research practices (reagent and protocol sharing, data management) and in part computer science (bioinformatic tools) as evidenced by the range of trainings offered so far. Tools we discuss could also be useful for disciplines close to biological research like bioengineering, biophysics, (bio)chemistry, etc. Some training modules, especially Data management, Data visualization and Figure design, might be

# Box 1. Unit topics.

The units included in the standard introductory workshop cover a range of skills and tools needed to conduct reproducible research. Below are examples of content that has been used in previous workshops. The specific content of each workshop can vary and is adjusted to the audience and event.

1. **The reproducibility framework:** Reproducible research practices allow others to repeat analyses and corroborate results from a previous study. This is only possible when authors have provided all necessary data, methods and computer codes (*Figure 2*). Our reproducibility toolbox includes reproducible practices for organization, documentation, analysis, and dissemination of scientific research.

2. **Organization, data management and file naming:** An effective data management plan, including clear file naming conventions, prevents problems such as lost data, difficulties identifying the most recent version of a file, the inability to locate files after team members leave the laboratory, or difficulties in finding or interpreting files years after the project is completed. This section describes techniques to ensure that all project files are easy to identify and locate and that they are appropriately documented.

3. **Electronic lab notebooks:** Electronic lab notebooks (ELNs) overcome many of the limitations of paper lab notebooks – they are searchable, cannot be damaged or misplaced, and are easy to back-up and share with collaborators. This section discusses available electronic lab notebooks and strategies for selecting the electronic lab notebook that meets the needs of an individual research team.

4. **Preregistrations and protocol sharing:** Scientific publications often lack essential details needed to reproduce the methods described. Preregistrations of planned research include details of the methods and tools that will be used in the project and provide transparency of the intended analyses and outcome. Protocol repositories allow researchers to share detailed, step-by-step protocols, which can be cited in scientific papers. Repositories also make it easy to track changes in protocols over time by incorporating version control, allowing researchers to post updated versions of protocols from their own lab, or adapted versions of protocols that were originally shared by other research groups. This section describes strategies for creating effective 'living protocols' that other research teams can easily locate, follow, cite and adapt.

5. **Biological material and reagent sharing:** Laboratories regularly produce specialized materials and organisms, such as reagents, plasmids, seeds and organism strains. Access to these materials is essential to reproduce and expand upon published work. Repositories maintain reagents and biological materials deposited by scientists, and also make these materials accessible to the scientific community for a small or symbolic donation. Nonetheless, many laboratories do not use repository resources to share their materials, and thus limit their outreach and impact. This section introduces the concept of material repositories, which allow investigators access to materials without investing time and resources to recreate, maintain, verify and distribute their own or another researcher's reagents.

6. **Data visualization and figure design:** Figures show the key findings of a study and should allow readers to critically evaluate the data and structures behind them. As an example, scientists routinely use the default plots of spreadsheet software such as bar graphs for presenting continuous data (*Weissgerber et al., 2015*). This is a problem, as many data distributions can lead to the same bar or line graph and the actual data may suggest different conclusions from the summary statistics alone. This section illustrates strategies for replacing bar graphs with more informative alternatives (i.e. dot, box, or violin plots), provides guidance on choosing the visualization best suited for various data structures and images, and provides a brief overview of tools for creating more effective, appealing and informative graphics and figures.

7. **Bioinformatic tools:** The sample size and number of data points (in multidimensional data) in research studies has greatly increased in the last decade. Bioinformatic tools for analyzing large data sets are essential in many fields. Unfortunately, analyses performed using these tools can only be reproduced or adapted to other study designs if authors share their code, software version and software settings. This section examines techniques and tools for reproducible data analysis, including notebooks, version control, managers for packages, dependencies and the programming environment, and containers.

8. **Data sharing:** Depositing data in public repositories allows other scientists to review published results and reuse the data for additional analyses or studies. All data should adhere to the principle of FAIR data: be findable, accessible, interoperable and reusable (https://fairsharing.org/). This section describes the types of information that should be shared to allow the community to interpret and use archived data. We also discuss best practices, including criteria for selecting a repository and the importance of specifying a license for data and code reuse. There are instances where data cannot be shared, this includes when there are privacy concerns with genetic data from living people.

## Box 2. Resources for training in reproducible research.

**Carpentries workshops** (https://carpentries.org/): Workshops teaching reproducible data handling and coding skills. Intended for scientists at any career stage.

**Frictionless Data Fellowship** (https://fellows.frictionlessdata.io): Nine-month virtual training program on frictionless data tools and approaches. Target audience are mainly early-career researchers (ECRs). Eight fellows are selected each year and a stipend is provided.

**Oxford Berlin Summer School** (https://www.bihealth.org/en/notices/oxford-berlin-summer-school-on-open-research-2020): Five-day summer school covering open research and reproducibility in science.

**ReproducibiliTea** (https://reproducibilitea.org/): Locally run journal clubs focused on open science and reproducibility. Target audience are mainly early career researchers. Global reach with currently 114 local groups.

**Research Transparency and Reproducibility Training** (RT2; https://bitss.org): Three-day training providing overview of reproducible tools and practices for social sciences. Target audience are scientists at any career stage of Social Sciences.

**Project TIER** (**Teaching Integrity in Empirical Research**) (https://www.projecttier.org/): Training in empirical research transparency and replicability for social scientists, students and faculty. Offer fellowships and workshops for faculty and graduate students.

**Framework for Open and Reproducible Research Training** (FORRT; https://forrt.org/): Connects educators and scholars in higher education to include open and reproducible science tenets in education. Offer the e-learning platform Nexus with several curated resources that include sufficient context for educators to use.

valuable for qualitative research that collects and analyzes text and other non-numerical data.

All materials, including recordings of previous R4E workshops and webinars, are available at https://www.repro4everyone.org/ (RRID:SCR_018958). The goal of R4E is to provide scientists with a clear overview of existing reproducibility-promoting tools, as well as to give scientists the opportunity to revisit all training material when needed, by providing them with full access to all training materials so they learn at their own pace. In addition, we welcome each trainee to fine-tune the material for their own field of expertise and to train their peers. For trainees who want to help run one of our workshops we offer the train-the-trainer approach: We meet with the trainee before the workshop and decide together which section of the material the trainee will present. Then we go through the material together, share speaker notes and practice with the trainee if needed to stay in time during the workshop.

We have developed materials for both introductory and intensive workshop formats that are described below:

- Introductory workshops are organized as two-hour sessions, including a 60- to 90 min presentation and 30 min interactive discussion of case studies, which can be held as in-person or virtual workshops with a large number of participants (>100). These introductory workshops are designed for an interdisciplinary audience and do not require prior knowledge of reproducible research practices as they cover many different topics (*Box 1*). These workshops are generally presented from a team of two to four instructors.

- Intensive workshops provide in-depth training in the implementation of reproducible research practices for one or more topics. These workshops take at least four hours. Depending on the number of topics covered, intensive workshops may be spread over several days. R4E members typically design these sessions to provide intensive instruction within their areas of expertise. Outside experts may also be invited to teach sessions on additional topics. This type of workshop is best suited for a smaller (<50) group of participants.

Over the years, our community has grown and diversified substantially, consisting of scientists who taught one, or many R4E workshops. To date, we have reached more than 3000 researchers through over 30 workshops, which were predominantly held at international conferences and spanned numerous life science disciplines (e.g. ecology, biotechnology, plant sciences, neuroscience and many others). In

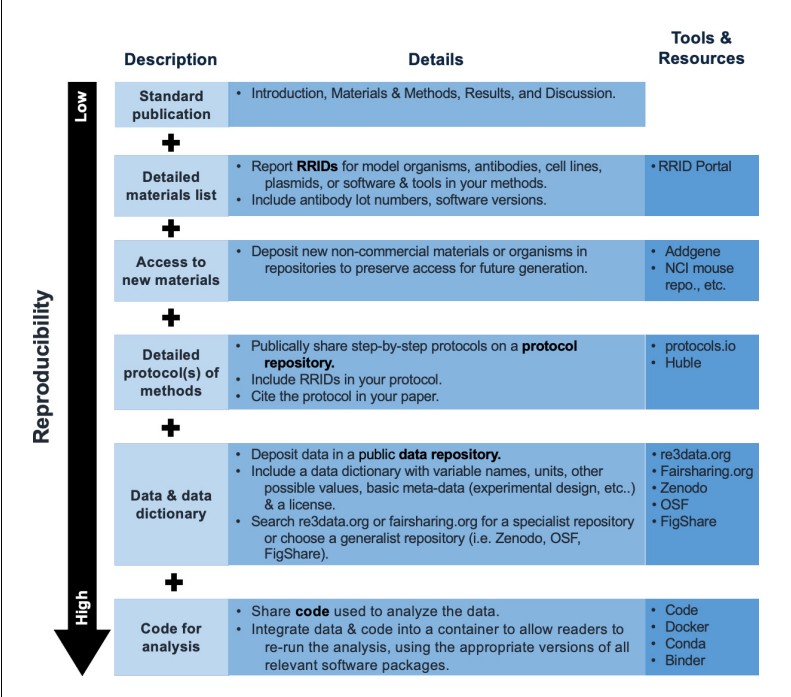

**Figure 2.** Approaches that scientists can use to increase the reproducibility of their publications. From top to bottom, approaches that can be used on their own or in combination to increase the reproducibility of experiments, ordered from least reproducibility to most. The column on the right includes details of tools and resources than can be used to help scientists take each specific approach.

addition, we have hosted several webinars that allowed researchers from all around the world to join, including webinars for early career scientists participating in the eLife Community Ambassadors Program. Investigators and conference organizers can request to host a workshop led by our volunteers or use our materials to learn more about responsible research practices and offer their own training.

The goal of our training is to introduce participants to a reproducible scientific workflow. Individual scientists or laboratories can make their research more reproducible by implementing as many of the steps introduced in our workshops as they are comfortable with (*Figure 2*). Feedback on our workshops indicate that 80% of participants learned important new aspects of reproducibly research practices and are very likely to implement at least some of the presented tools in their own research workflows in the future.

It is important to point out that this will likely work best as a stepwise, iterative process to avoid scientists from feeling overwhelmed with implementing too many changes at once. When writing a research paper, the largest impact on

the reproducibility of your work can be made by incorporating the following changes: adding a detailed list of materials used for the research, that includes research resource identifiers (RRIDs; https://scicrunch.org/resources) and catalog numbers for all materials (kits, antibodies, seeds, cell lines, organisms, etc.) that were created or used during the study. Ideally, newly generated reagents or organisms are deposited at appropriate repositories to enable easy access for other scientists. Incorporating a detailed and specific methods section is crucial to reproduce the research. Ideally, protocols are deposited at a repository, and the DOI number of the respective protocol is incorporated in the methods section. Large data sets, including all metadata, should be deposited in public data repositories to generate findable, accessible, interoperable, and reusable (FAIR) data (*Sansone et al., 2019*). Finally, bioinformatic analytic pipelines and scripts can easily be shared via Github, Anaconda, or computational containers such as Singularity. At a minimum, authors should list and cite all programs used, including version numbers and parameters.

We would also like to point out that a supportive environment is critical for these efforts to be properly adopted in a research environment. Being the first one to speak up about irreproducible research practices at your lab or institute can be challenging, or in some cases even isolating. In this case, getting involved with a local ReproducibiliTea journal club or reaching out to the initiative to start a chapter of your own can help you connect with like-minded individuals. Similarly, joining the R4E community and discussing these situations with our community members can help you find solutions to convince your peers and supervisors of the importance of incorporating reproducible research practices.

## How can scientists use the R4E materials?

There are several ways for researchers to take advantage of the materials presented here to teach reproducible research practices. First, researchers can request a workshop run by the Reproducibility for Everyone team for a conference via email (hello@repro4everyone.org). Alternatively, researchers can use the slides and training materials available on our website to organize their own workshops. Reproducibility can be integrated into the research curriculum by asking trainees to organize and run a poster workshop at an institutional or departmental

research day. Trainees can also discuss individual topics at journal clubs or as part of a methods course, after which they can develop plans to implement the identified solutions in their own research. Upcoming workshops and other opportunities to get involved and contribute will be shared through our Twitter account (@repro4everyone) and website (https://www.repro4everyone.org/).

## Conclusions

Widespread adoption of new tools and practices is urgently needed to make scientific publications more transparent and reproducible. This transition will require scalable and adaptable approaches to reproducibility education that allow scientists to efficiently learn new skills and share them with others in their lab, department and field.

R4E demonstrates how a common, public set of materials curated and maintained by a small group may form the basis for a global initiative to improve transparency and reproducibility in the life-sciences. Flexible materials allow instructors to adapt both the content and workshop format to meet the needs of the audience in their discipline. Continued training on reproducibility could be promoted in the laboratory by for instance changing every $n^{th}$ journal club to an educational meeting, discussing the latest developments in the reproducibility field.

Our workshops have reached over 3000 learners on six continents and continue to expand each year, offering a unique opportunity to train the next generation of scientists. Moving forward, R4E plans to broaden our reach by translating the existing materials into different languages and bring reproducibility training to more non-native English-speaking scientists. However, increasing training in reproducible research practices alone will not suffice to make all scientific findings reproducible. To achieve this goal, higher-level changes are needed to reduce the hypercompetitive nature of scientific research. Large structural and cultural changes are needed to transition from rewarding only breakthrough scientific findings, to promoting those that were performed using reproducible and transparent research practices.

## Acknowledgements

Members of the Reproducibility for Everyone initiative would like to thank all organizers, volunteers and staff who have helped over the years with running our workshops. We would like to thank the eLife Ambassador program, Addgene, Protocols.io, the American Society of Plant Biology, the American Society of Microbiology, New England Biolabs, the Chan-Zuckerberg Initiative, Dorothy Bishop, and many others for supporting the Reproducibility for Everyone initiative.

**Reproducibility for Everyone Team**
**Angela Abitua**: Addgene, Boston, United States; **Anzela Niraulu**: Neuroscience Graduate Program, Ohio State University, Columbus, United States; **Aparna Shah**: The Solomon H. Snyder Department of Neuroscience, Johns Hopkins University, Baltimore, United States; eLife ambassador, Cambridge, United Kingdom; **April Clyburne-Sherinb**: Reproducibility for Everyone, New York, United States; **Benoit Guiquel**: Addgene, London, United Kingdom; **Bradly Alicea**: Orthogonal Research and Education Laboratory, Champaign, United States; eLife ambassador, Cambridge, United Kingdom; **Caroline LaManna**: Addgene, Boston, United States; **Diep Ganguly**: Research School of Biology, Australian National University, Canberra, Australia; eLife ambassador, Cambridge, United Kingdom; **Eric Perkins**: Addgene, Boston, United States; **Helena Jambor**: Centre for Regenerative Therapies Dresden, Dresden, Germany; **Ian Man Ho Li**: Massachusetts General Hospital, Harvard University, Boston, United States; **Jennifer Tsang**: Addgene, Boston, United States; **Joanne Kamens**: Addgene, Boston, United States; **Lenny Teytelman**: Protocols.io, San, Francisco, United States; **Mariella Paul**: Psychology of Language Group, University of Gottingen, Gottingen, Germany; eLife ambassador, Cambridge, United Kingdom; **Michelle Cronin**: Addgene, Boston, United States; **Nicolas Schmelling**: Institute for Synthetic Microbiology, Heinrich-Heine-Universität Düsseldorf, Düsseldorf, Germany; **Peter Crisp**: Research School of Biology, Australian National University, Canberra, Australia; eLife ambassador, Cambridge, United Kingdom; **Rintu Kutum**: CSIR-Institute of Genomics and Integrative Biology, New Delhi, India; eLife ambassador, Cambridge, United Kingdom; **Santosh Phuyal**: Institute of Basic Medical Science, University of Oslo, Oslo, Norway; eLife ambassador, Cambridge, United Kingdom; **Sarvenaz Sarabipour**: Institute for Computational Medicine and Department of Biomedical Engineering, Johns Hopkins University, Baltimore, Baltimore, United States; Member of the eLife Early-Career Advisory Group, Cambridge, United Kingdom; eLife ambassador, Cambridge, United Kingdom; **Sonali Roy**: Plant Biotechnology, Tennessee State University, Nashville, United States; **Susanna M Bachle**: Addgene, Boston, United States; **Tuan Tran**: Aerospace Engineering, Nanyang Technological University, Singapore, Singapore; eLife ambassador, Cambridge, United Kingdom; **Tyler Ford**: Picture as Portal, San, Francisco, United States; **Vicky Steeves**: Research Data Management and Reproducibility, New York University, New, York, United States; **Vinodh Ilangovan**:

Max-Planck-Institut für biophysikalische Chemie, Göttingen, Germany; Member of the eLife Early-Career Advisory Group, Cambridge, United Kingdom; **Ana Baburamani**: Centre for the Development of Brain, School of Biomedical Engineering and Imaging Sciences, King's College London, London, United Kingdom; eLife ambassador, Cambridge, United Kingdom; **Susanna Bachle**: Addgene, Boston, United States;

**Susann Auer** is in the Department of Plant Physiology, Institute of Botany, Faculty of Biology, Technische Universität Dresden, Dresden, Germany and is an eLife ambassador

https://orcid.org/0000-0001-6566-5060

**Nele A Haeltermann** is in the Department of Molecular and Human Genetics, Baylor College of Medicine, Houston, United States and is an eLife ambassador

https://orcid.org/0000-0002-1431-7581

**Tracey L Weissgerber** is in the QUEST Center, Berlin Institute of Health, Charité Universitätsmedizin Berlin and is a member of the eLife Early-Career Advisory Group

https://orcid.org/0000-0002-7490-2600

**Jeffrey C Erlich** is in the NYU-ECNU Institute of Brain and Cognitive Science, NYU Shanghai and the Shanghai Key Laboratory of Brain Functional Genomics, East China Normal University, Shanghai, China and is an eLife ambassador

https://orcid.org/0000-0001-9073-7986

**Damar Susilaradeya** is in the Medical Technology Cluster, Indonesian Medical Education and Research Institute, Faculty of Medicine, Universitas Indonesia, Jakarta, Indonesia and is an eLife ambassador

https://orcid.org/0000-0002-4548-5924

**Magdalena Julkowska** is in the Boyce Thompson Institute, Ithaca, United States and is an eLife ambassador

**Małgorzata Anna Gazda** is in the CIBO/InBIOO, Centro de Investigaçao em Biodiversidade e Recursos Genéticos, Campus Agrário de Vairão and the Departamento de Biologia, Faculdade de Ciências, Universidade do Porto, Porto, Portugal and is an eLife ambassador

https://orcid.org/0000-0001-8369-1350

**Benjamin Schwessinger** is in the Research School of Biology, Australian National University, Canberra, Australia and is a member of the eLife Early-Career Advisory Group

benjamin.schwessinger@anu.edu.au

https://orcid.org/0000-0002-7194-2922

**Nafisa M Jadavji** is in the Department of Biomedical Science, Midwestern University, Glendale, United States and in the Department of Neuroscience, Carleton University, Ottawa, Canada and is an eLife ambassador

njadav@midwestern.edu

https://orcid.org/0000-0002-3557-7307

**Author contributions:** Susann Auer, Formal analysis, Visualization, Methodology, Writing - original draft, Writing - review and editing; Nele A Haeltermann, Małgorzata Anna Gazda, Reproducibility for Everyone Team, Visualization, Writing - original draft, Project administration, Writing - review and editing; Tracey L Weissgerber, Conceptualization, Funding acquisition, Visualization, Writing - original draft, Project administration, Writing - review and editing; Jeffrey C Erlich, Reproducibility for Everyone Team, Visualization, Writing - original draft; Damar Susilaradeya, Visualization, Writing - original draft, Project administration; Magdalena Julkowska, Reproducibility for Everyone Team, Visualization, Writing - original draft, Writing - review and editing; Benjamin Schwessinger, Reproducibility for Everyone Team, Conceptualization, Funding acquisition, Investigation, Methodology, Writing - original draft, Writing - review and editing; Nafisa M Jadavji, Supervision, Visualization, Writing - original draft, Project administration, Writing - review and editing

**Competing interests:** The authors declare that no competing interests exist.

**Funding**

| Funder | Grant reference number | Author |
| --- | --- | --- |
| Mozilla Foundation | MF-1811-05938 | Benjamin Schwessinger |
| Chan Zuckerberg Initiative | 223046 | Susann Auer Nele A Haeltermann Benjamin Schwessinger Nafisa M Jadavji Reproducibility for Everyone Team |

The funders had no role in study design, data collection and interpretation, or the decision to submit the work for publication.

**Decision letter and Author response**
Decision letter https://doi.org/10.7554/eLife.64719.sa1
Author response https://doi.org/10.7554/eLife.64719.sa2

## Additional files

### Data availability
No new data were generated in this study.

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
