## [Decision Letter]

Thank you for submitting your article "A community-led initiative for training in reproducible research" to *eLife* for consideration as a Feature Article. Your article has been reviewed by 3 peer reviewers, and the evaluation has been overseen by Helena Pérez Valle, and the *eLife* Features Editor, Peter Rodgers.

The reviewers and editors have discussed the reviews and we have drafted this decision letter to help you prepare a revised submission.

Summary:

This manuscript provides an overview of the community-led project Reproducibility for Everyone (R4E, https://repro4everyone.org/). R4E is already a useful training resource and community that provides introductory training sessions across a range of openness and reproducibility practices. Here the project is described so that researchers can make use of it to develop and teach reproducibility skills. This is a good collection of resources, and an excellent base collection for expansion via community engagement.

Essential revisions:

1. Please edit the first sentence of the abstract to temper it, for example, by being more specific about what conducting reproducible research does for the transparency and usefulness of research, and please avoid references to the scientific method in the article.

2. The article clearly defines replicable and reproducible separately but then uses the word reproducibility to encompass the things discussed in the paragraph from line 88-105 (lines are numbered lines from the PDF) which mainly describe issues with replicability. Please correct slippage of terminology throughout the article.

3. Please edit Figure 1 so that it is not presented as a Venn diagram (i.e. each factor should appear under just one heading, and the label "lack of training" should be removed) and edit the caption to describe the figure closely and explain that it is a rough approximation for the taxonomy of factors that contribute to research not being reproducible.

4. If possible, it would be informative for the article to include more statistics and comments about the reach and impact of the Reproducibility for All project. If any participant feedback or similar is available for sharing, please report it in the article. Information on whether the workshops are as successful when delivered virtually as they are in person would be particularly useful given the current context and the fact that scalability may rely on virtual formats.

5. Please consider adding a paragraph at an appropriate place in the paper covering the following comments made by the referees:

"Between the individual and systemic levels of science is a cultural level, including mentor and peer support, lab culture, and supervision. While the authors are rightly proud of their existing reach and have laudable ambitions for translating materials into different languages, they miss an opportunity to address cultural barriers. Including acknowledgement of and advice for researchers who are in unsupportive environments (or researchers who are pushing the reproducibility envelope for their institutions), with suggestions of where practical and social support can be found, would be a valuable inclusion. Addressing the isolation which can surround being the first in an institution to try to address reproducibility issues is, in my experience organising ReproducibiliTea Journal Clubs, all the more relevant for institutions outside the elite English-speaking universities."

"In my reading there is an implicit assumption throughout the manuscript that improving training and improving access to training should solve issues with reproducibility. I agree that training is part of the solution, but what this manuscript does not discuss in depth is grapple with the dynamics between for example improved training and pervasive incentives to not spend time on reproducibility. I do acknowledge that I may be reading too much into this aspect and that it is not the purpose of this manuscript to discuss the research culture issues around reproducibility other than to introduce R4E."

6. Please cite papers (or repositories of papers) that have enacted the practices summarised in Page 8.

7. Please include a discussion about the fields or kinds of research that the reproducibility pipeline refers to. It would be useful for the reader to know whether the approach presented by R4E was developed with different disciplines in mind, or a specific biological research focus that then needs to be ported elsewhere. A limitation of some of the 'open science' projects, for example, has been to narrow focus so much to a specific fields' needs or a specific research type that readers could presume that the approach was simply not valid for their research. R4E seems to cross this boundary nicely and this could be highlighted more in the manuscript. It may also be worth making a note of the applicability of these pipelines to very different research approaches, for example qualitative approaches.

---

## [Author Response]

Essential revisions:1. Please edit the first sentence of the abstract to temper it, for example, by being more specific about what conducting reproducible research does for the transparency and usefulness of research, and please avoid references to the scientific method in the article.

We thank the reviewer for this comment. As requested, we have revised the first sentence of the abstract to the following:

“Open and reproducible research practices increase the reusability and usefulness of scientific research.”

We have removed scientific method from the manuscript.

2. The article clearly defines replicable and reproducible separately but then uses the word reproducibility to encompass the things discussed in the paragraph from line 88-105 (lines are numbered lines from the PDF) which mainly describe issues with replicability. Please correct slippage of terminology throughout the article.

Thank you very much for this thoughtful reply. We discussed this issue and now added the following sentence to the manuscript on page 3, lines 76 to 77.

“We use the term reproducibility to capture both these concepts at once as is done often in life sciences (Barba, 2018).”

The provided reference is a good analysis that shows that different fields use these terms differently. We are of the view that in life sciences reproducibility is often understood at both terms interchangeably. We prefer to acknowledge these shortcomings as the article would become otherwise to complex and wordy as we often mean both reproducibility and replicability at the same time.

3. Please edit Figure 1 so that it is not presented as a Venn diagram (i.e. each factor should appear under just one heading, and the label "lack of training" should be removed) and edit the caption to describe the figure closely and explain that it is a rough approximation for the taxonomy of factors that contribute to research not being reproducible.

We have revised the figure so that the content is not presented as a Venn diagram. We have also revised the figure caption.

4. If possible, it would be informative for the article to include more statistics and comments about the reach and impact of the Reproducibility for All project. If any participant feedback or similar is available for sharing, please report it in the article. Information on whether the workshops are as successful when delivered virtually as they are in person would be particularly useful given the current context and the fact that scalability may rely on virtual formats.

We thank the reviewer for this comment, we have included the following statement within the manuscript (page 8, lines 199 to 201) as well as below.

“Feedback on our workshops indicate that 80% of participants learned important new aspects of reproducibly research practices and are very likely to implement at least some of the presented tools in their own research workflows in the future.”

We would like to point out that this is based on non-representative post-participation surveys, interviews with participants and anecdotal feedback. This is not based on a scientific approach or a fully controlled study. We hope the careful phrasing of the sentence make this clear.

Similarly, we have not seen a significant difference in the uptake comparing in person vs. online workshops. Yet again this has not been analyzed to the degree that we would feel comfortable commenting on this aspect in this manuscript. Hence, we feel this is out of the reach of the current manuscript that introduces R4E.

5. Please consider adding a paragraph at an appropriate place in the paper covering the following comments made by the referees:"Between the individual and systemic levels of science is a cultural level, including mentor and peer support, lab culture, and supervision. While the authors are rightly proud of their existing reach and have laudable ambitions for translating materials into different languages, they miss an opportunity to address cultural barriers. Including acknowledgement of and advice for researchers who are in unsupportive environments (or researchers who are pushing the reproducibility envelope for their institutions), with suggestions of where practical and social support can be found, would be a valuable inclusion. Addressing the isolation which can surround being the first in an institution to try to address reproducibility issues is, in my experience organising ReproducibiliTea Journal Clubs, all the more relevant for institutions outside the elite English-speaking universities."

We thank the reviewer for pointing this out to us and agree that it can be isolating and frustrating when you find yourself in a non-supportive environment. We have added the following to the manuscript to address this situation (page 9, lines 217 to 225):

“We would like to point out that a supportive environment is critical for these efforts to be properly adopted in a research environment. […] Similarly, joining the R4E community and discussing these situations with our community members can help you find solutions to convince your peers and supervisors of the importance of incorporating reproducible research practices.”

"In my reading there is an implicit assumption throughout the manuscript that improving training and improving access to training should solve issues with reproducibility. I agree that training is part of the solution, but what this manuscript does not discuss in depth is grapple with the dynamics between for example improved training and pervasive incentives to not spend time on reproducibility. I do acknowledge that I may be reading too much into this aspect and that it is not the purpose of this manuscript to discuss the research culture issues around reproducibility other than to introduce R4E."

We wholeheartedly agree with the reviewer’s comment: higher-level issues are to blame for scientific findings being irreproducible. However, as they are deeply engrained in our meritocracy-based science funding system, overcoming these barriers will require a complete re-organization of our current way of funding and evaluating scientific success. R4E tries to focus on the positive, and addresses changes that each scientist can implement themselves to make a difference in the reproducibility of their own research. If the reviewer would like to see this addressed in the manuscript, we could add the following section to the description of factors affecting reproducibility (Page 10, lines 259 to 262):

“Increasing training in reproducible research practices alone will not suffice to make all scientific findings reproducible. […] Large structural and cultural changes are needed to transition from rewarding only breakthrough scientific findings, to promoting those that were performed using reproducible and transparent research practices.”

6. Please cite papers (or repositories of papers) that have enacted the practices summarised in Page 8.

We are unfortunately unable to act on this request as we are unsure what the reviewer refers to. We think the reviewer request is too unspecific as given.

7. Please include a discussion about the fields or kinds of research that the reproducibility pipeline refers to. It would be useful for the reader to know whether the approach presented by R4E was developed with different disciplines in mind, or a specific biological research focus that then needs to be ported elsewhere. A limitation of some of the 'open science' projects, for example, has been to narrow focus so much to a specific fields' needs or a specific research type that readers could presume that the approach was simply not valid for their research. R4E seems to cross this boundary nicely and this could be highlighted more in the manuscript. It may also be worth making a note of the applicability of these pipelines to very different research approaches, for example qualitative approaches.

We thank the reviewers for this comment and have added a paragraph to bridge the gap and make it clearer that our training materials are not exclusively fitting for biological research only. We hope this improves the applicability of our materials. Some feedback we got during workshops from researchers outside of the life sciences indicates that our materials were useful for them as well (Page 6, lines 144 to 147).

R4E targets mainly biological and medical research practices (reagent and protocol sharing, data management) and in part computer science (bioinformatic tools) as evidenced by the range of trainings offered so far. Tools we discuss could also be useful for disciplines close to biological research like bioengineering, biophysics, (bio)chemistry, etc. Some training modules, especially Data management, Data visualization and Figure design, might be valuable for qualitative research that collects and analyzes text and other non-numerical data.